# LEARNING TO PROVE THEOREMS BY LEARNING TO GENERATE THEOREMS

## ABSTRACT

We consider the task of automated theorem proving, a key AI task. Deep learning has shown promise for training theorem provers, but there are limited human-written theorems and proofs available for supervised learning. To address this limitation, we propose to learn a neural generator that automatically synthesizes theorems and proofs for the purpose of training a theorem prover. Experiments on real-world tasks demonstrate that synthetic data from our approach significantly improves the theorem prover and advances the state of the art of automated theorem proving in Metamath.

## 1 INTRODUCTION

Automated theorem proving is a key task in Artificial Intelligence. The goal is to automatically generate a proof, given a conjecture (the target theorem) and a knowledge base of known facts, all expressed in a formal language. Automated theorem proving is useful in a wide range of applications, including the verification and synthesis of software and hardware systems (Gu et al., 2016; Darvas et al., 2005; Kern & Greenstreet, 1999).

Automated theorem proving boils down to a search problem: finding the sequence of symbol manipulations that generate a valid proof. A prover typically works backward: starting from the theorem statement, it searches for a path that connects the theorem to known facts in the knowledge base. The fundamental challenge lies in the explosion of search space, in particular with long proofs and large knowledge bases. The success of theorem proving thus relies on effective heuristics that guide the search by deciding the next step the prover should take.

Deep learning has emerged as a promising approach to learning search heuristics in a automated theorem prover (Irving et al., 2016; Yang & Deng, 2019; Whalen, 2016; Loos et al., 2017; Bansal et al., 2019a). The search process fundamentally reduces to a sequence of actions on manipulating a set of symbols. Thus a deep network can be trained to select the best action at each step.

A key challenge is how to train such networks. Prior work has used human-written theorems and proofs to perform imitation learning and has shown promising results (Loos et al., 2017; Yang & Deng, 2019; Whalen, 2016; Paliwal et al., 2019). The training data consists of theorems and proofs manually written by human experts in a formal language, and the prover is trained to imitate the proof steps demonstrated by humans.

However, relying on human-written data has a major drawback, that is, such data has limited availability and scalability. Writing theorems and proofs in a formal language requires highly specialized knowledge and skill, including mathematics, computer programming, and proficiency in the particular formal language. For a computer science graduate student, it can take months to master a new formal language such as Mizar, Metamath or HOLight (Wiedijk, 2003), after which it can take days to formalize a single page of a math textbook. This makes it impractical to crowdsource human-written proofs at large scale.

An alternative to imitation learning is reinforcement learning, which requires only formalized theorem statements but not their proofs. During training, the prover estimates the value of each action through exploration. This reinforcement learning approach substantially reduces the amount of manual formalization needed, but at the expense of sample efficiency. The prover needs positive rewards to assess past attempts, but positive rewards are only available when the prover finds a com-

plete proof, which is rare because it involves a combination of multiple correct steps. This leads to extremely sparse positive rewards, and in turn very low sample efficiency.

In this paper, we propose to learn search heuristics using synthetic data. The basic idea is to construct a generator that automatically synthesizes new theorems and their proofs, which are then used to augment human-written data. To generate a new theorem and its proof, the generator applies an inference rule on a set of existing theorems and combines their proofs to form the proof of the new theorem. Similar to the prover, the generator performs a sequence of symbol manipulations, albeit in the inverse direction, going forward from existing theorems to a new theorem instead of from a target theorem to existing ones.

A key question is how to construct a generator such that the generated data is useful. The space of new theorems and proofs is infinite, but a prover can only process a finite amount of data during training. Thus, to maximize the utility of the generate data, we make the generator learnable by parametrizing it with deep networks.

We hypothesize that the generated data will be more useful if they are similar to human-written data. Thus we use human-written data to train a generator. We consider two scenarios. If the human-written data consists of both theorem statements and their proofs, we train the generator to follow the proof steps in the forward direction, so that a well-trained generator would derive theorems humans tend to derive. If the human-written data consists of only theorem statements but not their proofs, we use reinforcement learning to train the generator such that the generated theorems are similar to the human-written theorems. We measure similarity using the language model trained on the human-written theorem.

We instantiate our approach in Metamath (Megill, 2019), a popular language for formal mathematics, and with Holophrasm (Whalen, 2016), a Metamath neural prover. We propose a neural theorem generator we call "MetaGen", which synthesizes new theorems and their proofs expressed in the formalism of Metamath. To the best of our knowledge, MetaGen is the first neural generator of synthetic training data for theorem proving. Experiments on real-world Metamath tasks demonstrate that synthetic data from MetaGen helps the prover prove more human-written theorems, achieving state of the art results. Experiments also show that our approach can synthesize useful data, even when there are only human-written theorems but zero proofs during training.

## 2 RELATED WORK

**Automated theorem proving** Our work is related to prior work on learning to prove theorems (Whalen, 2016; Gauthier et al., 2018; Bansal et al., 2019a; Yang & Deng, 2019; Loos et al., 2017; Balunovic et al., 2018; Kaliszyk et al., 2018; Bansal et al., 2019b). Our work directly builds off of Holophrasm (Whalen, 2016), a neural-augmented theorem prover for Metamath. It contains three deep networks to generate actions and initial values to guide proof search following the UCT algorithm (Kocsis & Szepesvári, 2006).

TacticToe (Gauthier et al., 2018), DeepHOL (Bansal et al., 2019a) and ASTactic (Yang & Deng, 2019) are learning-based theorem provers for higher-order logic based on various interactive theorem provers, including HOL4 (Slind & Norrish, 2008), HOL Light (HOLLight) and Coq (Bertot & Castéran, 2004). Paliwal et al. (2019) improves DeepHOL by representing formulas as graphs. Loos et al. (2017) propose to learn clause selection by deep learning inside the first-order logic prover E (Schulz, 2002). FastSMT (Balunovic et al., 2018) learns to compose search heuristics as programs with branches for the SMT solver (De Moura & Bjørner, 2008).

All of these methods are othogonal to our approach because all of their provers are learned from human-written training data, whereas our contribution is on training a neural generator of synthetic training data for theorem proving.

Kaliszyk et al. (2018); Bansal et al. (2019a;b) use reinforcement learning to train provers with only human-written theorems but not their proofs. During training, a prover only collects rewards only upon finding full proofs. In contrast, we always train our prover using imitation learning. Under the same setting with only human-written theorems but not proofs, we use reinforcement learning to train our generator, whose reward is the similarity between a generated theorem and a human-written theorem, as measured by a language model of human-written theorems. Our reinforcement

learning task is much easier because the reward is continuous and there are many ways to generate theorems similar to human-written ones.

**Automatic goal generation by self-play** Our work is similar to the line of work in reinforcement learning (Florensa et al., 2018; Sukhbaatar et al., 2017; 2018; Durugkar & Stone, 2018) that deploys one agent to generate tasks for another agent to accomplish. Sukhbaatar et al. (2017); Florensa et al. (2018) propose to train these two agents by adversary self-play, where the generation agent learns to produce difficult goals for another agent. With self-play, the generator learns to increase the difficulty of goals and build a learning curriculum automatically.

We pursue similar ideas in the new context of theorem proving by learning to generate synthetic theorems to train the prover. Also of note is that we have no adversarial self-play. The goal of the generator is to discover novel theorems similar to human-written ones, not to beat the prover.

Recently, Huang (2019) introduced a two-player game which encourages players to learn to predict the consistency of formulas in first-order logic by self-play. These two players behave symmetrically and complete with each other in the game. In contrast, our generator and prover execute different tasks, and are co-operative. In addition, their game remains a theoretical proposal without any empirical validation, whereas we have performed experiments on large-scale data.

## 3 BACKGROUND ON METAMATH

Metamath is a language for developing formal mathematics. It is one of the simplest formal systems. It has only one inference rule, called *substitution*, but is universally applicable in formalizing a large portion of mathematics [1] and different types of logic (Megill, 2019).

A knowledge base in Metamath consists of a set of theorems including axioms, which are admitted to be true, and others that are derived from proofs. Each theorem has one or more expressions, including one assertion and zero or more hypotheses. The hypotheses provide the preconditions, such as $x = y^2$ and $y$ is an even number, to prove the assertion, such as $x$ is divisible by 4.

Following Whalen (2016), an expression is represented as a tree of tokens, whose nodes are either constants or variables. A constant node has a fixed number of children (including zero) and a variable has no children. Therefore, we represent each expression as a unique sequence of tokens by traversing its parse tree in pre-order.

A proof is a sequence of steps using substitution. A proof step has two parts, a theorem that is declared earlier than the current theorem in the knowledge base, and a substitution that maps a variable in this theorem to a new expression. For example, we have a theorem $t$,

$$\text{hypotheses:} \qquad A = B \tag{1}$$
$$\text{assertion:} \qquad CFA = CFB \tag{2}$$

$\{A, B, C, F\}$ is the set of variables in $t$. Let $\phi$ be a substitution to map each variable in $t$ to a new expression,

$$A \to 2 \qquad B \to (1 + 1) \qquad C \to 2 \qquad F \to + \tag{3}$$

By replacing variables in the $t$ with their corresponding expressions from $\phi$, we have a new assertion and a set of new hypotheses,

$$\text{new hypotheses:} \qquad 2 = (1 + 1) \tag{4}$$
$$\text{new assertion:} \qquad 2 + 2 = 2 + (1 + 1) \tag{5}$$

and this proof step $(t, \phi)$ demonstrates that the new assertion $2 + 2 = 2 + (1 + 1)$ is entailed by the new hypothesis $2 = (1 + 1)$. Note that we need to substitute all occurrences of the same variable with the same expression in both the assertion and hypotheses [2].

Formally, let $e \in \mathcal{E}$ be an expression (a tree of tokens) with $l$ unique variables $\mathbf{f}_e = (f_1, f_2, ..., f_l) \in \mathcal{F}^l$, and let $\phi \in \mathcal{F} \to \mathcal{E}$ be a substitution. Let $e(\phi)$ denote the new expression obtained by replacing

---

[1] Its largest knowledge base, `set.mm` ranks 3rd in the "Formalizing 100 Theorems" challenge (Wiedijk).

[2] Variables in Metamath are called metavariables, which are different from variables bound by quantifiers in the first-order and higher-order logic.

$f_i$ in $e$ with $\phi(f_i)$, for $i = 1, 2..., l$. And given $k$ expressions $\mathbf{e} = (e_1, e_2, ..., e_k) \in \mathcal{E}^k$, let $\mathbf{e}(\phi) = (e_1(\phi), e_2(\phi), ..., e_k(\phi))$ represent applying the substitution to all $k$ expressions.

Given a theorem $t$, let $a_t$ be its assertion and $\mathbf{h}_t = (h_{t,1}, h_{t,2}, ..., h_{t,m}) \in \mathcal{E}^m$ be its hypotheses. Let $\phi_t$ be a substitution for variables in $t$. A proof step $s = (t, \phi_t) \in \mathcal{E}^{m+1} \times (\mathcal{F} \to \mathcal{E})$ demonstrates an entailment of assertion $a_t(\phi_t)$ by hypotheses $\mathbf{h}_t(\phi_t)$.

In this formal system, proving a theorem $\tau$ means finding a tree such that (1) the root node is the assertion of $\tau$, (2) each leaf node is either empty or one of the hypotheses of $\tau$, and (3) each internal node is an expression associated with a proof step that demonstrates an entailment of the node by its children.

To prove a target theorem $\tau$, it is the most straightforward to reason backward. We start by selecting a proof step that will demonstrate an entailment of the assertion of the target theorem, that is, $a_t(\phi) = a_\tau$. It is worth noting that if we pick a particular theorem $t$, if a valid $\phi$ exists, $\phi(f)$ is uniquely determined for any variable $f$ that occurs in the assertion (recall that each expression is a tree of tokens). But $\phi(f)$ is not uniquely determined if $f$ only occurs in the hypotheses ($f$ is called a *hypothesis variable*, or a *assertion variable* if it only occurs in the assertion), because it can be re-placed with anything. Once this initial proof step $(t, \phi)$ is fully specified, the assertion $a_\tau$ is entailed by a set of new hypotheses $\mathbf{h}_\tau(\phi)$, and the goal of proving theorem $\tau$ has now been decomposed to the subgoals of finding entailment of the new hypotheses $\mathbf{h}_\tau(\phi)$ by the original hypotheses $\mathbf{h}_\tau$.

## 4 APPROACH

### 4.1 PROBLEM STATEMENT AND APPROACH OVERVIEW

In our task setup, we assume two sets of pre-existing human-written theorems. There is a set of "background theorems", $B = \{b_1, b_2, ..., b_n\}$, which can be freely used as known facts in both training and testing. Each proof step consists of a background theorem $b \in B$ and its substitution.

There is also a set of "target theorems", $T = \{t_1, t_2, .., t_n\}$, which are theorems to prove. The target theorems are split into a training set, a validation set, and a test set. The target theorems in the test are not seen during training. A prover learns to prove the target theorems in the training set. The final goal of the prover to perform well on the target theorems in the test set.

For target theorems in the training set and the validation set, a subset of them have human-written proofs. Let $S = (\mathbf{s}_1, \mathbf{s}_2, ..., \mathbf{s}_k)$ be proofs for $k$ target theorems $(t_{s,1}, t_{s,2}, ..., t_{s,k}) \subseteq T$ and $\mathbf{s}_i = (s_i^1, s_i^2, ...s_i^{l_i})$ be a sequence of proof steps for $i = 1, 2, ..., k$. We also consider the case with only theorems but zero proofs.

Our approach consists of two modules, a prover to find proofs for given target theorem and a generator to sample synthetic theorems along their proofs. We call these synthetic theorems as "generated theorems". These two modules performs similarly but in opposed directions. They both generate proof steps iteratively to find a proof for the target theorem or generate new theorems.

The prover executes backward reasoning. Given a target theorem, it builds the proof search tree where a node represents a goal. It starts from the assertion as the root and extends the proof search tree by decomposing current goals using new proof steps. It succeed if a complete proof is found.

The generator runs in forward reasoning. It maintains a graph of theorems where nodes in the graph are all theorems it has encountered during training, and each edge represents entailment in a single proof step between a precondition and the assertion. It samples new proof steps using existing theorems as preconditions and add the generated theorems to the graph. Generated theorems are then used to train the prover.

For both the prover and the generator, the policies to pick a proof step are parameterized by deep networks with similar architectures but different inputs.

We first train the generator using target theorems and optionally their proofs. Then sample synthetic proofs and train the prover from both human proofs and synthetic proofs.

**Algorithm 1:** MetaGen

**Input:** $B\ T\ S\ N$
/* thms targets proofs #steps */
**Variable:** $V$, $E$
/* nodes edges                  */

**Function** Generate():
  Initialize()
  **while** $V.length \leq N$ **do**
    /* A new proof step.      */
    Sample $b \sim D_1$                // (**)
    $\phi, \mathbf{q} \longleftarrow \emptyset, \emptyset$
    **for** $h \in \mathbf{h}_b$ **do**
      /* Sample a precondition
        node for $h$          */
      **try**
        Sample $q, \phi_q \sim D_2$
      **except** *Failed*
        Restart from (**)
      **end**
      Add $q$ to $\mathbf{q}$
      Merge $\phi_q$ to $\phi$
    **end**
    **for** $f \in b$'s assertion variables **do**
      /* Generate a
        substitution for $f$ */
      **try**
        Sample $e_f \sim D_3$
      **except** *Failed*
        Restart from (**)
      **end**
      $\phi(f) \longleftarrow e_f$
    **end**
    AddNode($b$, $\phi$, $\mathbf{q}$)
  **end**
  **return**
**end**

**Function** Initialize():
  $V, E \longleftarrow \emptyset, \emptyset$
  **for** $t \in T$ **do**
    **for** $h \in \mathbf{h}_t$ **do**
      /* Add a hypothesis   */
      Add $(h, \{h\})$ to V
    **end**
  **end**
  **for** $\mathbf{s} \in S$ **do**
    **for** $(b, \phi) \in \mathbf{s}$ **do**
      /* Add a step $(b, \phi)$     */
      $\mathbf{q} \longleftarrow$ *precondition nodes* for $(b, \phi)$
      AddNode($b$, $\phi$, $\mathbf{q}$)
    **end**
  **end**
  **return**
**end**

**Function** AddNode($b$, $\phi$, $\mathbf{q}$):
  $a, \mathbf{h} \leftarrow a_b(\phi), \emptyset$
  **for** $q \in \mathbf{q}$ **do**
    /* Merge hypotheses      */
    Merge $\mathbf{h}_q$ to $\mathbf{h}$
  **end**
  $v \leftarrow (a, \mathbf{h})$
  **if** $v \notin V$ **then**
    /* A new node.           */
    Add $v$ to $V$
    **for** $q \in \mathbf{q}$ **do**
      Add $(q, v)$ to $E$
    **end**
  **end**
  **return**
**end**

## 4.2 GENERATOR

### 4.2.1 GENERATION PROCEDURE

The generator maintains a graph of theorems $G = (V, E)$, where $V = H \bigcup Q$ is the set of nodes and $E$ is the set of edges.

$H$ is the set of hypotheses, which is collected from target theorems $T$ as $H = \bigcup_{t \in T} \mathbf{h}_t$. $H$ is fixed and we never generate new hypotheses. For convenience, we write a hypothesis $h$ as $(h, \{h\})$ in the same form as a theorem, such that we have $a_h = h$, $\mathbf{h}_h = \{h\}$.

$Q = \{(a_1, \mathbf{h}_1), (a_2, \mathbf{h}_2), (a_3, \mathbf{h}_3), ...\}$ is the set of theorems that have been encountered by the generator. Here we record each node in the form of theorems instead of assertions, because hypotheses play an important role to the meaning and the proof of a theorem. We hope to keep track of different proofs for the same assertion from different hypotheses.

$G$ is a directed graph. An edge $(u, v) \in E$ represents that $v$ is entailed in a proof step that uses $a_u$ the assertion of $u$ as one of the preconditions. A hypothesis node has no incoming edges. For a node, if multiple proofs exist, we preserve the edges from the last proof step of its shortest proof.

To generate a new theorem, a proof step is sampled such that every precondition for this step is identical to the assertion of a node in the graph. We call this node the *precondition node*.

Formally, given a proof step $s = (b, \phi)$ and *precondition nodes* $\mathbf{q} \subseteq V$, $s$ demonstrates that $a_b(\phi)$ is entailed by the preconditions $\mathbf{h}_b(\phi)$. For $h \in \mathbf{h}_b(\phi)$, we have a node $q_h \in \mathbf{q}$ such that $a_{q_h} = h$. Then we merge hypotheses from *precondition nodes* to get $\mathbf{h} = \bigcup_{h \in \mathbf{h}_b(\phi)} \mathbf{h}_{q_h}$ and compose a theorem $(a_b(\phi), \mathbf{h})$, which means $a_b(\phi)$ can be proved from the new hypotheses $\mathbf{h}$ merged from *precondition nodes*. It is added to $Q$ if it is different from other nodes in the graph. This procedure is summarized as `AddNode` in Algorithm 1.

Initially, we add hypotheses from target theorems $T$ into $H$.

If we have at least one training proof, assume $t_{\mathbf{s}} \in T$ is the target theorem for a proof $\mathbf{s} = (s^1, s^2, ..., s^l) \in S$. For a proof step $s^i = (b, \phi) \in \mathbf{s}$, where $i = 1, 2, ..., l$, each precondition $h \in \mathbf{h}_b(\phi)$ for $s^i$ is either a hypothesis of $t_{\mathbf{s}}$ or an intermediate assertion that is entailed in a previous step $s^j \in \mathbf{s}$, where $j < i$. This means a node $q$ satisfying $a_q = h$ has been added to the graph $G$ previously. So we can find *precondition nodes* for $s^i$ and add this step to the graph.

The above process is also summarized as `Initialize` in Algorithm 1.

After initializing the graph $G$ from training data, we sample proof steps upon nodes in $G$ as *precondition nodes* to produce new generated theorems. We repeat this procedure until we obtain all the generated theorems in our expectation to training the prover.

To sample a proof step, we need to make three decisions sequentially.

- Sample a background theorem $b \sim D_1(B)$.
- Given an empty substitution $\phi$ and *precondition nodes* $\mathbf{q}$, for each hypothesis $h_{b,i} \in \mathbf{h}_b$, sample a node $q_i \sim D_2(b, \phi, \mathbf{q})$ and its corresponding substitution $\phi_i$, such that $h_{b,i}(\phi_i) = a_{q_i}$ and $\phi_i$ is consistent with the existing substitution $\phi$. Update $\phi$ and $\mathbf{q}$ using $\phi_i$ and $q_i$.
- Optionally, if the theorem $b$ has *assertion variables*, for each assertion variable $f$, sample an expression $e \sim D_3(f, b, \phi, \mathbf{q})$ and update $\phi$.

The distribution $D_1(B)$ is either uniform over $T$ or multinomial over the frequencies of background theorems used in training proofs.

$D_2$ is a distribution over all nodes in $G$ that can result in a substitution $\phi_i$ consistent with $\phi$ and $h_{b,i}$. $D_2$ is multinomial over softmax probabilities outputted from a deep network, called the relevance network following Whalen (2016).

Graph $G$ could have millions of nodes. This makes it intractable to compute $D_2$ over all nodes in $G$. In practice, we limit the number of nodes for $D_2$ to compute. Given a hypothesis $h$, to sample a precondition node, we retrieve all nodes that could fit $h$ by substitution. If we get too many candidates, we sample 2000 nodes randomly and calculate $D_2$ over these 2000 candidates using the relevance network.

$D_3$ is a distribution over potential expressions. We parameterize $D_3$ by a sequence generation network, which produces the pre-order traversal sequences for the expression trees. We call this network as substitution network instead of generation network used in Whalen (2016) to prevent it from getting confused with the generator.

If we cannot find a feasible *precondition node* for a hypothesis of $b$ or generate a legal expression to an assertion variable, we fail in this iteration and restart by sampling a theorem.

The complete generation procedure is summarized as `Generate` in algorithm 1. The architecture of the relevance and substitution networks of the generator is presented in Appendix A.1.1.

### 4.2.2 TRAINING OF THE GENERATOR

We propose two training strategies to learn the generator, with or without training proofs.

With access to training proofs, we train the generator using Imitation Learning. After initializing the graph $G$ from training proofs, the generator is trained to imitate each proof step $s = (b, \phi)$. For each $h \in \mathbf{h}_b$, we use a node $q$ where $a_q = h(\phi)$ as supervision for the relevance network to learn

to pick precondition nodes. For each $b$'s assertion variable $f$, we use the $\phi(f)$ as supervision for the substitution network to learn to generate substitution for $f$.

We call the generator trained in this strategy as *MetaGen-IL*.

If we have no access to training proofs, we train the generator using Reinforcement Learning. We first learn a language model from target theorems using a GRU. We expect this language model assigns low cross-entropy to generated theorems that are similar to target theorems. Then we use this language model as a reward function to train the generator with the Reinforce algorithm. We run the generator to sample new generated theorems on-the-fly. New theorems are evaluated by the cross-entropy of the language model and the generator is updated toward lower cross-entropy.

We call the generator trained in this strategy as *MetaGen-RL*.

### 4.3 PROVER

We adopt Holophrasm as our theorem prover. Given a target theorem, Holophrasm tries to find the proof by backward reasoning. It starts from the assertion and builds the proof search tree that consists of goals and proof steps. Each goal has multiple proof steps as its children. Each proof step $(b, \phi)$, which demonstrates that $a_b(\phi)$ is entailed by $\mathbf{h}_b(\phi)$, has $a_b(\phi)$ as its parent and a set of subgoals $\mathbf{h}_b(\phi)$ as its children. A proof step is solved if it has no children or all of its children are solved. A goal is solved if at least one of its proof steps is solved. The proof is found when the assertion in the root is solved.

Holophrasm maintains valuation for goals and proof steps. It explores the proof search tree using UCT (Kocsis & Szepesvári, 2006). The search direction is controlled by the valuation, which is dependent on a node's initial value and visit count following the UCB criterion.

The initial value of a goal is from a payoff network. To generate a proof step for the current goal, a relevance network picks a theorem and optionally a substitution network produces substitutions for *hypothesis variables*. The initial value of this step is also calculated using the prediction scores from relevance and substitution networks.

All deep networks (relevance, substitution, payoff) of the prover are trained by imitation learning. We mix the human proofs for training theorems and synthetic proofs generated by MetaGen and optimize deep networks to make the same decision as a training proof step. Noted that, we don't limit the number of candidates for relevance networks in the prover as we do in the generator.

The architecture and more training details of deep networks for the prover is presented in Appendix A.1.2.

## 5 EXPERIMENTS

### 5.1 EXPERIMENTAL SETUP

To evaluate our proposed methods, we train the prover on the mixture of human proofs (if available) and synthetic proofs sampled by our proposed generator *MetaGen*, in three different settings: (1) with 10% training proofs, (2) with all training proofs, and (3) with no training proofs.

**Dataset** We use the Metamath `set.mm` knowledge base as our dataset. `set.mm` formalizes the Tarski-Grothendieck set theory. It contains 29337 theorems and they are all used as background theorems $B$ in algorithm 1. We remove axioms from $B$ and use the rest of theorems as target theorems $T$ that are divided into three sets, including 21788 training theorems, 2712 validation theorems and 2720 test theorems.

The generator is trained on training theorems and their human proofs optionally. For experiments with no training proofs, we learn a language model from training theorems and use it as the reward function to train MetaGen-RL. For other two experiments with 10% or all human proofs of training theorems, we train MetaGen-IL on the given proofs as described in section 4.2.2.

**Baselines** For all experiments, we compare with Holophrasm trained on human proofs only by imitation learning the same as Whalen (2016).

Table 1: The performance of relevance networks on validation data evaluated by the average probability and top-K accuracy to pick positive theorems.

| Human proofs | Synthetic proof steps | Generator | Model | Prob | Top-1 | Top-5 | Top-20 |
|---|---|---|---|---|---|---|---|
| 0 | 0 | - | tf-idf | 0.0081 | 27.82 | 36.15 | 47.68 |
| 0 | 0 | - | relevance | 0.0065 | 0.279 | 1.924 | 12.50 |
| 0 | 200K | *MetaGen-RL* | relevance | 0.2160 | 22.23 | 42.23 | 60.54 |
| 2179 | 0 | - | relevance | 0.6007 | 62.14 | 76.30 | 89.30 |
| 2179 | 1M | *MetaGen-IL* | relevance | 0.5889 | 62.14 | 76.29 | 87.61 |
| 21788 | 0 | - | relevance | 0.5978 | 61.54 | 74.55 | 87.28 |
| 21788 | 10M | *MetaGen-IL* | relevance | 0.5920 | 63.02 | 76.71 | 87.93 |

Table 2: The performance of substitution networks on validation data evaluated by the average probability and accuracy to generate the ground truth token at each position with teacher forcing.

| Human proofs | Synthetic proof steps | Generator | Model | Prob | Accuracy |
|---|---|---|---|---|---|
| 0 | 0 | - | lauguage model | 0.0032 | 9.06 |
| 0 | 0 | - | substitution | 0.0008 | 0.01 |
| 0 | 200K | *MetaGen-RL* | substitution | 0.0050 | 25.03 |
| 2179 | 0 | - | substitution | 0.2738 | 58.91 |
| 2179 | 1M | *MetaGen-IL-Rand* | substitution | 0.3203 | 61.78 |
| 2179 | 1M | *MetaGen-IL* | substitution | 0.3710 | 66.56 |
| 21788 | 0 | - | substitution | 0.6142 | 81.57 |
| 21788 | 10M | *MetaGen-IL* | substitution | 0.6847 | 83.90 |

With 10% training proofs, we also compare with a random generator baseline *MetaGen-IL-Rand*.

With no human proofs, we propose also an ad-hoc baseline by replacing relevance and substitution networks with TF-IDF similarities and a naive language model. For more details on baseline methods, please refer to Appendix A.2.

**Evaluation**   The prover is evaluated by the number theorems proved on the test set. Following the settings used in Whalen (2016), the prover is run on each test target theorem for 5 minutes or through 10000 passes, in three runs with different beam search width as 1,5,20 for substitution networks.

Since payoff networks are trained on exclusive data sampled by relevance and substitution networks, they can not be compared among different methods. In general, we found payoff networks work similarly, with accuracy ranging from 78 to 82.

Please also refer to Appendix A.3 for our full implementation details and hyper-parameter settings.

We present the performance of the relevance and substitution networks and the final provers in the next section.

## 5.2   RESULTS

**Relevance network**   We evaluate relevance networks by the average probability and top-K accuracy to pick positive theorems among all feasible candidates. From table 1, we got similar performance for all networks trained using human proofs, even among those trained on 10% or all proofs.

Table 3: The number of theorems proved on test data.

| Human proofs | Synthetic proof steps | Generator | Prover | Test proofs found |
|---|---|---|---|---|
| 0 | 0 | - | tf-idf $\&$ LM | 312 |
| 0 | 0 | - | Holophrasm | 219 |
| 0 | 200K | *MetaGen-RL* | Holophrasm | 351 |
| 2179 | 0 | - | Holophrasm | 452 |
| 2179 | 1M | *MetaGen-IL-Rand* | Holophrasm | 461 |
| 2179 | 1M | *MetaGen-IL* | Holophrasm | 475 |
| 21788 | 0 | - | Holophrasm('16) | 388 |
| 21788 | 0 | - | Holophrasm | 539 |
| 21788 | 10M | *MetaGen-IL* | Holophrasm | 574 |

Table 4: The examples of generated theorems for *MetaGen-IL* trained on all human proofs.

| Hypothesis | Assertion | Comment |
|---|---|---|
| $\emptyset$ | $(3 \times 1) + (1 + 0) = 1 + 3$ | Simple arithmetic. |
| $\emptyset$ | $(\log e) \times A = A$ | $e = 2.71828...$ |
| $A \in \mathbb{C}$ $B \in \mathbb{C}$ | $\sin(A + B) = (\exp(\mathbf{i} \times (A + B))$ $- \exp(-\mathbf{i} \times (A + B)) \div (2 \times \mathbf{i})$ | $\mathbb{C}$: complex number set. $\mathbf{i} = \sqrt{-1}$. |
| $\emptyset$ | $G \in \mathbb{R} \wedge E \in \mathbb{R} \to \sin(\frac{G+E}{2} + 1) \in \mathbb{R}$ | $\mathbb{R}$: real number set. |
| $\phi \to F \colon X \leftrightarrow Y$ | $\phi \to \mathrm{Ran}(F) \subseteq Y$ | F: bijection from X to Y. $\mathrm{Ran}(F)$: range of $F$. |
| $N = \{x \in \mathbb{Z} | M \le x\}$ | $\phi \wedge K \in N \to$ $M \in \{x \in \mathbb{Z} | M \le x \wedge x \le K\}$ | $\mathbb{Z}$: integer set |
| $r = q \times 2 \times y \mod p$ $s = q \times 2 \times x \mod p$ | $x = y \to F(r \times y) = F(s \times x)$ | mod: modulo operation |
| $x \in \mathrm{Base}(g)$ | $g \in \mathrm{Grp} \wedge x \in \mathrm{Fin} \wedge p \in \mathrm{Prime} \wedge$ $h \in \mathrm{pSyl}(p, g) \to h \in \mathrm{SubGrp}(g)$ | Base: base extractor set Grp: all groups Fin: all finite sets pSyl: Sylow p-subgroup |

So the size of training data isn't the bottleneck to improve the performance of relevance networks. Therefore, we also have no room of boost by generating synthetic proofs.

We also found the relevance network trained on synthetic proofs sampled by *MetaGen-RL* performs better than the TF-IDF baseline, especially in terms of average probability, which is directly related to the valuation maintained by the prover. It means even *MetaGen-RL* is trained without proofs, it still samples synthetic proofs that are helpful for the relevance network.

**Substitution network** As shown in table 2, we use the accuracy and probability to generate the correct token under teacher forcing as metrics to compare different substitution networks.

In all settings, the performance of substitution networks are improved by a large margin by using synthetic data. This demonstrates the synthetic proofs generated by *MetaGen* are effective in training substitution networks.

With 10% proofs, *MetaGen-IL-Rand* beats the baseline trained on 10% proofs only. This result demonstrates the robustness of the *MetaGen* algorithm even without learned parameters. *MetaGen-IL* outperforms *MetaGen-IL-Rand* and this verifies that the learning of the generator is effective.

**Prover** Since we cannot promote relevance networks using synthetic proofs, we only apply relevance networks trained on real proofs only, to study if improvement from substitution networks could be reflected on provers.

As shown in table 3, provers augmented with synthetic proofs generated by *MetaGen* proved more target theorems than all baseline methods with the same amount of human proofs.

Our re-implementation of Holophrasm baseline using all proofs proves 539 test theorems, which is much higher than reported by Whalen (2016). We assume it is due to our GPU implementation, which brings the speed advantages since all provers run in the same time limit.

Given no training proofs, the original Holophrasm performs worst since it carries two randomly initialized networks. After training on synthetic proofs generated by *MetaGen-RL*, Holophrasm proves 140 more test theorems. It also find 12.5% more proofs than TF-IDF and language model baseline. This shows the effectiveness of *MetaGen-Rl*.

By incorporating *MetaGen-IL* with all training proofs, we prove 574 theorems on the `set.mm` knowledge base, which is the new state-of-the-art result on this benchmark.

**Generated theorems** We show some examples of the generated theorems in table 4. Generated theorems range over different topics, such as basic arithmetic, complex analysis, basic function operation, number theory and algebraic structures.

Some theorems provide important and interesting results, such as the third one, a equation involving the trigonometric function and the complex formula. Some are boring for human, such as the first one and the fourth one, but they also provide meaningful training data for the prover. The first one takes five steps to prove and the fourth one takes four steps. Although their proofs consist of common used theorems, such as commutative law of addition and real field is closed to arithmetic, they provide novel scenarios to exercise these theorems. Without humans' intuition and abstraction, such "trivial" exercises are helpful for deep networks to strengthen their memorization and generalize to broader range of theorems.

## 6 CONCLUSION

We have proposed a neural generator that automatically synthesizes theorems and proofs for the purpose of training a theorem prover. Experiments on real-world tasks have demonstrated that synthetic data from our approach significantly improves the theorem prover and advances the state of the art of automated theorem proving in Metamath.

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

# A  APPENDIX

## A.1  NETWORK ARCHITECTURE

Our deep networks follow the same architecture as Holophrasm(Whalen, 2016). We represent each expression as a sequence of tokens. Both the relevance and substitution networks take such expression sequences as inputs. A embedding layer is learned for tokens in the vocabulary and GRUs (Cho et al., 2014) are used to encode and decode input sequences.

### A.1.1  GENERATOR

The relevance network picks a *precondition node* among some theorem nodes according to the background theorem $b$ and previously selected *precondition nodes*. It uses a theorem encoder to embed the background theorem $b$ as concatenation of the assertion and hypotheses of $b$ and hypotheses of selected *precondition nodes*. It uses another precondition encoder to embed each theorem node as concatenation of its assertion and hypotheses. These two embeddings are fed to a bilinear layer for the score of this node. The softmax probabilities are outputted over all candidates.

The substitution network is a sequence-to-sequence model. It generates the substitution for a variable $f$ in background theorem $b$, given a set of *precondition nodes*. It replaces $f$ as special token to mark itself as the target, then uses a encoder-decoder GRU to generate the output sequence from concatenation for the assertion of $b$ and hypotheses of *precondition nodes*.

### A.1.2  PROVER

The relevance network picks a theorem on a goal given a proof task. It has a theorem encoder to embed each potential theorem as the concatenation of its assertion and hypotheses. It also has a goal encoder to embed the concatenation of goal and the hypotheses of the task. A bilinear layer and the softmax function follows these two encoders to output the probability over potential theorems.

The substitution network is a sequence-to-sequence model. After replacing a target variable as a special token, it takes inputs as the concatenation of hypotheses of the task and hypotheses of a picked theorem, passes it through a encoder-decoder to generate a sequence tokens as the substitution.

The payoff network predicts if a goal can be proved in the given proof task. So it takes the concatenation of the goal and hypotheses of the task as inputs and use a GRU encoder and two linear layers to get the score for the current goal.

The relevance and substitution networks are learned from training proofs. Given a proof step $s = (b, \phi)$, the relevance network learns to predict $b$ from other theorems and the substitution network learns to generate $\phi(f)$ for a hypothesis variable $f$ in $b$.

The payoff network works as a binary classifier of intermediate goals. Positive instances are extracted as goals from training proofs. Negative instances are generated using the learned relevance and substitution networks. For each goal in training proofs, we generate proof steps and its subgoals. The subgoals not covered by the proof are treated as negative instances.

## A.2 BASELINE

For all experiments, we use the prover trained on human proofs only as baselines.

This leads to a prover carrying randomly initialized networks when we have no training proofs.

With 10% training proofs, we also compare *MetaGen-IL* with a random generator *MetaGen-IL-Rand*, where $D_1, D_2$ in algorithm 1 are uniform and $D_3$ is uniform over substitutions occured in training proofs to verify if the learning of the generator is helpful.

With no human proofs, we also manually design an ad-hoc baseline to replace the randomly initialized relevance and subsitution networks. We pick a theorem over the cosine similarities between TF-IDF features of theorems and goals, which is used in Bansal et al. (2019b), to replace the relevance network. We learn a language model which is trained on expressions among target theorems, including assertions and hypotheses, to generate new expressions as substitutions and replace the substitution network. We compare our *MetaGen-RL* method with this baseline.

## A.3 IMPLEMENTATION DETAILS

The dimensions of all GRUs are 128 except the language model for *MetaGen-RL*, which is 64-dimensional. All deep networks are trained using Adam ,where the learning rate is 1e-4 for relevance and payoff networks, and 5e-4 for substitution networks, in both generators and provers. Learning rates are always decreased by 2 in the manually-scheduled epochs. The batch size of all networks are 100.

To train relevance networks in the generator/prover, we need to discriminate a generated theorem/background theorem from the other candidates, whose number could be higher than 10K. During training, we sample 10 negative candidates for each positive theorem in one iteration. During inference, generators sample one target theorem from at most 20 candidates and provers pick each theorem from all feasible candidates.

Substitutions are produced as sequences of tokens. For generators, each token is sampled over probabilities from substitution networks. For provers, substitutions are generated by beam search.

The networks in *MetaGen-IL* for the first two experiments are trained for 6000 iterations each epoch. With 10% training proofs, we train networks in *MetaGen-IL* for 40 epochs and decrease learning rates after 20, 26 and 32 epochs. With all training proofs, we train networks in *MetaGen-IL* for 60 epochs and decrease learning rates after 30, 40 and 50 epochs.

To train *MetaGen-RL*, we train the language model for 200 epochs and decrease learning rates after 80, 120 and 160 epochs. Both the relevance and substitution networks are trained by Reinforce for 20 epochs. In each epoch, we reinitialize the generated theorem graph and sample 8000 new unique proof steps. Networks are updated for each 40 theorems. We set the gradient clipping as 5 to stabilize policy gradients.

With learned generators, we sample synthetic proofs following the algorithm 1. The distribution $D_1$ is multinomial over frequencies of theorems used by training proofs in *MetaGen-IL*, and uniform over all theorems in *MetaGen-RL*. With 10% training proofs, we sample about 1M unique proof steps and therefore the theorem graph is extended into 1M nodes. With all training proofs, we run *MetaGen-IL* 10 times and sample 1M unique proof steps in each run and then merge them as 10M synthetic proofs. With no training proofs, we sample 200K unique proof steps using *MetaGen-RL*.

Given all synthetic data, we train deep networks in the provers. With access to training proofs, we mix synthetic proof steps and real proof steps in each batch, in ratio of 7:3 for 10% training proofs and in ratio of 5:5 for all training proofs. With no training proofs, we train the prover using synthetic data only.

Following the settings in Whalen (2016), the prover is run on each test target theorem for 5 minutes or through 10000 passes. The beam search width is 1, 5, 20 in three different runs.

