# OpenReview forum: "Learning to Prove Theorems by Learning to Generate Theorems"
_ICLR.cc/2020/Conference — Reject_

### Official Review · AnonReviewer3 · 2019-10-23
**Official Blind Review #3**

**Rating:** 6

**Review:**

This paper proposes a generative model for proofs in Metamath, a language for formalizing mathematics. The model includes neural networks, which provide guidance about which fact to try to prove next and how to prove the fact from the facts derived so far. The parameters of these networks are learned from existing proofs or theorem statements. The main purpose of this model is to generate synthetic theorems and proofs that can be used to train the neural networks of a data-driven search-based theorem prover. The experiments with the Metamath set.mm knowledge base show the benefits of the synthetically generated proofs for building a data-driven theorem prover.

I think that the paper studies an important problem and contains interesting ideas. The idea of using a language model for theorem statements (so that a generated theorem can be meaningfully compared with a given theorem even when they are not the same) looks sensible. Also, the conjecture that a good proof generator is likely to lead to a good theorem prover sounds plausible.

I find the description of the training of the generative model in the experiments slightly confusing. Adding some clarification may help some readers. More specifically, here are some questions that I couldn't answer for myself. What theory is formalized by set.mm? Set theory? Among the proofs of 29337 theorems, which ones are used during the training of the generative model?


Here are some minor comments.

* p1: positive awards ===> positive rewards

* p2: A citation is missing in the first sentence of Section 2.

* AddNode, Algorithm1, p5: Merge h_q to h' ===> Merge h_q to h

* p6: uses a_v as a precondition ===> uses a_u as a precondition

* p6: and has been ===> has been

* p7: which demonstrate ===> which demonstrates

* p7: from these the relevance ===> from the relevance

* p7: wiht ===> with

* p9: languagee ===> language


**Experience Assessment:**

I have read many papers in this area.

**Review Assessment: Checking Correctness Of Derivations And Theory:**

N/A

**Review Assessment: Checking Correctness Of Experiments:**

I assessed the sensibility of the experiments.

**Review Assessment: Thoroughness In Paper Reading:**

I read the paper thoroughly.

---

> ### Author Response · Authors · 2019-11-15
> **Response to reviewer#3**
>
> Thank you for your comments and your time for reviewing our submission. We address your questions below.
>
> Q1: What theory is formalized by set.mm? Set theory?
>
> A: Set.mm formalizes the Tarski-Grothendieck set theory. We added this information to the second paragraph of section 5.1 in the revision.
>
> Q2: Among the proofs of 29337 theorems, which ones are used during the training of the generative model?
>
> A: The same set used to train the prover.
>
> We conducted experiments in three settings to train the generator with zero human proofs, 10% human proofs or all human proofs from all proofs of the target theorems in the training set.
>
> Q3: minor comments
>
> A: Thanks! We have addressed them in our revision.

---

### Official Review · AnonReviewer1 · 2019-11-01
**Official Blind Review #1**

**Rating:** 6

**Review:**

This paper focuses on the task of automated theorem proving. To address the low availability of human-written data and low sample efficiency in reinforcement learning, the authors propose to augment data by generating synthetic theorem data with a deep neural network-based model. Experimental results show the usefulness of the generated synthetic theorem.

This paper is well-motivated and the proposed method is quite novel for automated theorem proving. The paper is well-supported by theorems, however, the experimental analysis is a little weak. For the above reasons, I tend to accept this paper but wouldn't mind rejecting it.

Questions:
1. Maybe it's better if you can shorten section 3 and explain more about the problem setting (such as how to fit this problem in a graph?).
2. Can you show some examples of generated theorems?
3. You showed the prover has better performance with more synthetic data, but why is your model (generator) better? Can other generative models generate better proofs?

**Experience Assessment:**

I have read many papers in this area.

**Review Assessment: Checking Correctness Of Derivations And Theory:**

I assessed the sensibility of the derivations and theory.

**Review Assessment: Checking Correctness Of Experiments:**

I assessed the sensibility of the experiments.

**Review Assessment: Thoroughness In Paper Reading:**

I read the paper at least twice and used my best judgement in assessing the paper.

---

> ### Author Response · Authors · 2019-11-15
> **Response to reviewer#1**
>
> Thank you for your comments and your time for reviewing our submission. We address your questions below.
>
> Q1: Maybe it's better if you can shorten section 3 and explain more about the problem setting (such as how to fit this problem in a graph?).
>
> A: We revised section 4.1 and 4.2.1 and added more clarification.
>
> Q2: Can you show some examples of generated theorems?
>
> A: The following examples of generated theorems are shown in table 4 and discussed in the last two paragraphs of section 5.2 in our revision.
>
> Assertion:
>     ( ( 3 * 1 ) + ( 1 + 0 ) ) = ( 1 + 3 )
>
> Assertion:
>    ( ( log e ) * A ) = A    // e is Euler's constant 2.71828….
>
> Hypothesis:
>    A \in CC, B \in CC // x \in y means “x belongs to y”. CC is the complex number  set.
> Assertion:
>    sin ( A + B ) = ( exp ( i * ( A + B ) ) - exp ( ( - i ) * ( A + B ) )  ) / ( 2 * i )   //  i is the square root of -1.
>
> Assertion:
>    ( G \in R /\ E \in R ) -> ( sin ( ( G + E ) / 2 ) + 1 ) \in R
> // R is the real number set.
>
> Hypothesis:
>    phi -> F : X -1-1-onto-> Y   // F is a bijective mapping from X to Y.
> Assertion:
>    phi -> Ran F C_ Y    // the range of F is a subset of Y.
>
> Hypothesis:
>    N = { x \in Z | M <= x }
> Assertion:
>    ( phi /\ m \in N ) -> M \in { x \in Z | M <= x /\ x <= N }
>
> Hypothesis:
>    R = ( Q * 2 * y ) mod P   //mod is module operation
>    S = ( Q * 2 * x ) mod P
> Assertion:
>    x = y -> F ( R * y ) = F ( S * x )
>
> Hypothesis:
>    X \in Base(G) // X is a base extractor of G.
> Assertion:
>    ( G \in Group ) /\ ( X \in FiniteSet ) /\ ( P \in PrimeNumber) /\ ( H \in Sylow P-subgroup(G, p) ) -> ( H \in SubGroup(G) )
>
> Q3: You showed the prover has better performance with more synthetic data, but why is your model (generator) better? Can other generative models generate better proofs?
>
> A: To the best of our knowledge, MetaGen is the first generative model for theorems, so we are not aware of alternative models for comparison. Generative models developed for other domains such as images or texts are not directly applicable because theorem generation must comply with strict symbolic rules that generative models of images or natural texts do not need to handle.

---

### Official Review · AnonReviewer4 · 2019-11-02
**Official Blind Review #4**

**Rating:** 3

**Review:**

This paper focuses on the problem of developing deep learning systems that can prove theorems in a mathematical formalism -- in this case, MetaMath. This has been a rapidly growing topic in the past few years, as evidenced by the numerous cited works. What sets this work apart from others is its focus on the instrumental task of generating data to train a prover, rather than directly training the prover on human theorems (via reinforcement learning) or human proofs (via imitation learning).

The paper proposer two approaches to generating theorems imitation learning (IL) and reinforcement learning (RL). The IL approach trains a neural policy to imitate the same steps taken in human proofs. The RL approach first trains a language model on human theorems (not proofs), and uses the likelihood under the model as a reward function for an RL agent which must take forward proof steps.

Both approaches result in a policy that can be used to take proof steps, with the goal of producing new theorems which are similar to the human ones. Since the proof steps are known for the generated theorems, a prover agent (which operates in backwards mode, working from the goal back to the hypotheses) can be trained to imitate the steps taken in the synthetic proofs (along with the human ones, if any are present).

At test time, the learned prover imitation policy is then used to guide an MCTS agent, as described in the Holophrasm paper. It is compared against the original Holophrasm algorithm, rerun on modern hardware.

This is to my knowledge a novel approach in the neural theorem proving domain, and in my opinion one that offers a potentially significant advantage over the existing fixed-dataset appraoches.

The main result of the paper is that an extra 35/2720 (1.2%) of the test theorems are proven, a 6% improvement over the Holophrasm baseline of 539. It is difficult to judge how relevant of an improvement this is, and there is no analysis of the difficulty of the MetaMath problem set. In addition, due to the 10-1-1 train-validation-test split, the neural agents are likely shown relatively similar problems during training as at test time, including potentially stronger versions of the same theorems. There is also no comparison against non-neural approaches, such as Z3, Vampire, or similar theorem provers.

To accept this paper, I would like to see stronger evidence that the introduced method produces significant improvements in prover ability. For example, the same method could be applied to datasets such as HOList, Mizar, and CoqGym which have received more attention recently than MetaMath.

Some additional questions and comments:
1. How big does the theorem graph G get? Since the relevance policy is over all nodes of the graph, this could lead to a very large neural network that would be difficult to fit into memory. Certainly not all 1M synthetic theorems could be generated in one graph.
2. The paper claims that all theorems from set.mm are used as background theorems in algorithm 1, including the test ones -- this potentially sounds like training on the test set, or even worse, having access to the test theorems as "proven background knowledge" at test time.
3. Please include some more details about the training of the Holophrasm baseline. Does it simply do RL on the human theorems, or does it also do IL on human proofs?

**Experience Assessment:**

I have read many papers in this area.

**Review Assessment: Checking Correctness Of Derivations And Theory:**

I carefully checked the derivations and theory.

**Review Assessment: Checking Correctness Of Experiments:**

I carefully checked the experiments.

**Review Assessment: Thoroughness In Paper Reading:**

I read the paper thoroughly.

---

> ### Author Response · Authors · 2019-11-15
> **Response to Reviewer#4**
>
> Thank you for your comments and your time for reviewing our submission. We address your individual points below in a QA format.
>
> Q1: The main result of the paper is that an extra 35/2720 (1.2%) of the test theorems are proven, a 6% improvement over the Holophrasm baseline of 539. It is difficult to judge how relevant of an improvement this is, and there is no analysis of the difficulty of the MetaMath problem set.
>
> A: In our experiments, the improvement from MetaGen over the Holophrasm baseline is significant because it is virtually impossible to prove a new theorem by random guessing. The average proof length is 55 in set.mm, and the prover can find a proof only after taking a long sequence of correct proof steps. In addition, a proof step can require composing a new expression, further increasing the search space. This means that the probability of proving a new theorem through random guessing is close to zero, and proving a few dozens more theorems is a significant improvement. As shown in Table 3, we achieve consistent improvement from MetaGen in different training settings. When trained on all human proofs, our method with MetaGen-IL could find 21 extra proofs with five proof steps or more.
>
> Q2: The same method could be applied to datasets such as HOList, Mizar, and CoqGym which have received more attention recently than Metamath.
>
> A: Set.mm in Metamath is a good benchmark for automated theorem proving. Mathmath only relies on substitution, the most general and fundamental inference rule of deductive reasoning, and therefore can serve as a meta-language to implement different logics, like first-order logic, higher-order logic, and set theory, while other systems are usually built on a particular logical foundation. Such simplicity and generality offer a unique advantage for developing ML provers, because we can generate all potential theorems by handling substitution only.
>
> Set.mm is the largest corpus of math theorems in Metamath. It contains 29,337 theorems and almost 1.5M proof steps. It implements the Tarski-Grothendieck set theory and covers various math topics, including but not limited to first-order logic, real and complex analysis, linear algebra, graph theory, elementary geometry and topology. It formalizes 71 of the “top 100” math theorems, only behind HOL Light and Isabelle/HOL among all formal math databases [1] , and its coverage is still actively growing. This makes set.mm a good benchmark to train and evaluate learning-based theorem provers.
>
> The idea of theorem generation can be applied to other systems beyond Metamath, but realizing it on another system is highly nontrivial. It can even involve new research challenges. In particular, due to large differences in logic foundations, grammar, inference rules, and benchmarking environments, the generation process, which is a key component of our approach, would be almost completely different for a new system. And the entire pipeline essentially needs to be re-designed and re-coded from scratch for a new formal system, which can require an unreasonable amount of engineering. Because of this, it is a standard practice in prior work to target a specific formal system and experiment only in this system [2,3,4,5,6,7,8].
>
> In addition, existing benchmarking environments for other systems have limitations that make it infeasible to implement our method. HOList [2] and CoqGym [3] are built on tactic-based theorem provers. Their environments only provide interfaces to call tactics implemented in backend provers. Most tactics execute backward reasoning. To generate new theorems, we need to be able to execute the corresponding reverse tactics, but this functionality is not provided in the current version of HOList and CoqGym.
>
> Our approach cannot be directly applied to Mizar, because it does not provide human proofs in a format that can be understood by an automatic prover like the E prover (see [5]). Prior works have used machine learning to improve the E prover [4,5,6] on Mizar, but they have only trained on proofs automatically found by the E prover, not those written by humans. E expresses theorems as CNFs and proves by refutation at the level of CNF clauses. The CNF representation of theorems and proofs are incomprehensible to humans. Thus it is an open research question how to do forward reasoning to generate synthetic theorems in the CNF form that are similar to human theorems.

---

> > ### Author Response · Authors · 2019-11-15
> > **Response to reviewer#4**
> >
> > Q3: There is also no comparison against non-neural approaches, such as Z3, Vampire, or similar theorem provers.
> >
> > A: Our main claim is that generating synthetic training data improves a learned prover. Comparison with non-learning provers does not validate or invalidate our claim.
> >
> > That said, we agree that it would still be informative to compare with traditional provers. However, traditional theorem provers like Z3, E and Vampire can not be directly applied to set.mm. Besides set theory, set.mm is also based on the theory of class and distinct variable provisos that are not used in Z3, E and Vampire. Adapting these provers to set.mm would be a research question on its own, and we are not aware of any existing work in this direction.
> >
> > Some interactive theorem provers (ITP) have hammers tools which translate theorems expressed in the ITP language into proper inputs for traditional theorem provers,  such as Z3, E and Vampire, and call these provers to prove the translated theorems. Such hammer tools include Sledgehammer [7] for Isabelle/HOL, HOLyHammer [8] for HOL light and MizAR for Mizar [5]. But to the best of our knowledge, no such tools exist for Metamath, and developing them would be a research topic on its own.
> >
> > Q4:  Due to the 10-1-1 train-validation-test split, the neural agents are likely shown relatively similar problems during training as at test time, including potentially stronger versions of the same theorems.
> >
> > A: We use 8-1-1 train-validation-test split in order to fairly compare to prior work [8], which uses the same split. Our main claim is that using synthetic data helps a learned prover. A 8-1-1 split, which gives ample training data to the baseline, would in fact better validate our claim than a 6-2-2 split, because a 6-2-2 split would provide less training data to the baseline, making the task more difficult for the baseline. This may in fact give more advantage to our approach, which generates synthetic data to address the lack of natural training data.
> >
> > In addition, our experiments have included the settings of using 10% and 0% of the human proofs, which provide even less natural training data than a 6-2-2 split. Our results show that our method gives consistent improvement over the baseline.
> >
> > Regarding seeing similar problems in training, set.mm consists of classical theorems that are formalized manually; due to the heavy labor involved, it is extremely rare to see redundant or near-duplicate theorems or proof steps. In addition, if Theorem A serves as a lemma for Theorem B, the proof of B just directly uses the conclusion of A without repeating the proof of A. So seeing the proof of A does not help to prove B, and vice versa.
> >
> > Q5: How big does the theorem graph G get? Since the relevance policy is over all nodes of the graph, this could lead to a very large neural network that would be difficult to fit into memory. Certainly not all 1M synthetic theorems could be generated in one graph.
> >
> > A: Our largest graph G has about 460K nodes from all human proofs and another 1M nodes from synthetic proofs. For relevance policy of the generator, the number of candidate nodes is limited to 2000. It means we sample 2000 nodes randomly if there are too many nodes fitting the current hypothesis. Therefore we can generate all 1M synthetic theorems in one graph.  In the revision, we have added more details about how we limit the number of candidates for the relevance policy of the generator in the last fourth paragraph of section 4.2.1.

---

> > > ### Author Response · Authors · 2019-11-15
> > > **Response to reviewer#4**
> > >
> > > Q6: The paper claims that all theorems from set.mm are used as background theorems in algorithm 1, including the test ones -- this potentially sounds like training on the test set, or even worse, having access to the test theorems as "proven background knowledge" at test time.
> > >
> > > A: Our setup is a standard one that has been used by many prior works [2,4,5,6]. And this is not “training on the test set”. Seeing a test theorem during training merely means seeing the *statement* of the theorem, not its proof. When a test theorem is used as background knowledge, it just means that the statement of the theorem is assumed as a known fact like an axiom. It doesn’t mean that the prover is told how to prove the theorem. For example, we can be given a proof of a theorem assuming the Riemann hypothesis, but this proof does not teach us how to prove the Riemann hypothesis. And the Riemann hypothesis can still be used as a theorem to be proved during test time.
> > >
> > > Q7: Please include some more details about the training of the Holophrasm baseline. Does it simply do RL on the human theorems, or does it also do IL on human proofs?
> > >
> > > A: The Holophrasm baseline is trained on human proofs by imitation learning the same as prior work [8]. We have added this information in our revision.
> > >
> > > [1] http://www.cs.ru.nl/~freek/100/
> > > [2] Kshitij Bansal, Sarah Loos, Markus Rabe, Christian Szegedy, and Stewart Wilcox. Holist: An environment for machine learning of higher order logic theorem proving. In International Conference on Machine Learning, 2019a.
> > > [3] Kaiyu Yang and Jia Deng. Learning to prove theorems via interacting with proof assistants. In International Conference on Machine Learning, 2019.
> > > [4] Geoffrey Irving, Christian Szegedy, Alexander A Alemi, Niklas Ee ́n, Franc ̧ois Chollet, and Josef Ur- ban. Deepmath-deep sequence models for premise selection. In Advances in Neural Information Processing Systems, 2016.
> > > [5] Cezary Kaliszyk and Josef Urban. MizAR 40 for Mizar 40. arXiv preprint arXiv:1310.2805
> > > [6] Sarah Loos, Geoffrey Irving, Christian Szegedy, and Cezary Kaliszyk. Deep network guided proof search. arXiv preprint arXiv:1701.06972, 2017.
> > > [7] Paulsson, Lawrence C., and Jasmin C. Blanchette. Three years of experience with Sledgehammer, a practical link between automatic and interactive theorem provers.
> > > [8] Whalen, Daniel. "Holophrasm: a neural automated theorem prover for higher-order logic." arXiv preprint arXiv:1608.02644(2016).

---

### Author Response · Authors · 2019-11-15
**Summary of our revision**

We thank all reviewers for their helpful comments! We revised our paper accordingly as follows.

1. We added examples of the generated theorems in table 4 and corresponding discussion in the last two paragraphs of section 5.2.

2. We added a paragraph to clarify the training of the generative model in the third paragraph of  section 5.1.

3. We added explanations on how we limit the number of candidate nodes for relevance networks of the generator in the last fourth paragraph of section 4.2.1.

4. We updated the section 4.1 and 4.2.1 to clarify the problem setting and the construction of the theorem graph.

---

### Public Comment · ~David_A_Wheeler1 · 2020-01-16
**Interesting work! Should be published.**

Note: I am not on the conference program committee, I am instead an interested bystander. I do have some connections with this paper that I believe I must make clear. I am a co-author of the cited book on Metamath, and I also have a long-standing background in AI. In any case, I hope my comments are helpful.

This paper should be published. The idea of using generators to improve machine learning is absolutely not new.
However, performing an experiment to actually applying this approach in the area of completely general-purpose unconstrained theorem provers *is* new.

I disagree with the ICLR 2020 Conference Program Chairs' decision. The paper *is* tailored to one specific formal system (Metamath), but that is completely *necessary* today. Different formal systems are quite different, and it is unreasonable to expect any researchers to re-implement multiple massive systems to perform a single experiment. There is ongoing work to try to bridge these systems to allow interoperation, but until those efforts are ready (if they ever are), performing experiments using specific formal systems is the only way to make progress in this area given current research funding levels.

Their results are interesting. Simply re-executing Holophrasm with better hardware shows a remarkable improvement
(from 388 to 539). Their extensive work here produced a surprisingly modest additional gain, from 539 to 574 in the best case. That said, it is still a gain in a hard area, and it also demonstrates the challenges of the approach they've taken. Science needs not just papers that document spectacular improvements; it also needs to report how "obvious" approaches provide more modest improvements or even make things worse (especially if it appears plausible that the results would have been spectacular). This paper provides an important data point for those trying to improve ML-based systems to prove mathematical theorems.

Here are my more specific comments.

References: the reference to "Metamath: A Computer Language for Mathematical Proofs" of 2019 lists Norman Megill's name, but it omits "David A. Wheeler" (the co-author). I hope you'll correct that :-).

Abstract: Change "we propose to learn" to "we propose to train". Also, I would remove "significantly"; it's a modest improvement, but it's a modest improvement in a hard area and that is nothing to be ashamed of.

Page 3: The definition of "theorem" here is different from the way it is used in the Metamath community (where it only refers to provable assertions). The term is used consistently in the paper, so I wouldn't change it, but it might be worth noting that difference here to reduce confusion.

Section 5: The setup section says that once axioms were removed there were 21788 training thoerems, 2712 validation theorems, and 2720 training theorems. Those are exactly the same numbers as Holophrasm. Can I assume that you used exactly the same version of the set.mm database? If so, that should be clearly stated, as that makes it much clearer that you are keeping things constant in your experiment (which is good!).

It is my sincere hope that the code will soon be released with an open source software (OSS) license so others can replicate and build on this work. After all, this work builds on Holophrasm, which was released on GitHub as open source software. I searched on GitHub and https://paperswithcode.com/paper/learning-to-prove-theorems-by-learning-to but didn't find it. Please do so!

Here are several easily-fixed nits:

Page 2: Change "their provers are learned from" to "their provers are trained from"
Page 2: Remove the first duplicate "only" in "a prover only collects rewards only".
Page 4: Change "along their proofs" to "along with their proofs"
Page 4: Change "These two modules performs" to "These two modules perform"
Page 4: Change "samples... and add the generated" to "samples... and adds the generated"
Page 4: remove the extraneous "the" in "it is the most straightforward to reason backwards".
Page 4: Change "Then sample" to "Then we sample"
Page 7: Change "Noted that," to "Note that" in "Noted that, we don't limit the number...".
Page 7: Change "For other two experiments" to "For the other two experiments"
Page 9: "no room of boost" isn't grammatical, that should be fixed.
Page 9: Change "It means even MetaGen-RL" to "It means that even if MetaGen-RL"
Page 10: Change "It also find" to "It also finds"

Perhaps an editor could quickly check for missing articles (a/an/the), singular/plural agreement, and verb conjugation throughout the paper. These are easily fixed, and they are common problems (especially for non-native speakers). They should be fixed for clarity and so that these nits don't detract from the work here.

---

### Decision · Program_Chairs · 2019-12-19

**Decision:**

Reject

**Comment:**

This paper proposes to augment training data for theorem provers by learning a deep neural generator that generates data to train a prover, resulting in an improvement over the Holophrasm baseline prover. The results were restricted to one particular mathematical formalism -- MetaMath, a limitation raised one by reviewer.

All reviewers agree that it's an interesting method for addressing an important problem. However there were some concerns about the strength of the experimental results from R4 and R1. R4 in particular wanted to see results on more datasets, an assessment with which I agree. Although the authors argued vigorously against using other datasets, I am not convinced. For instance, they claim that other datasets do not afford the opportunity to generate new theorems, or the human proofs provided cannot be understood by an automatic prover. In their words,

"The idea of theorem generation can be applied to other systems beyond Metamath, but realizing it on another system is highly nontrivial. It can even involve new research challenges. In particular, due to large differences in logic foundations, grammar, inference rules, and benchmarking environments, the generation process, which is a key component of our approach, would be almost completely different for a new system. And the entire pipeline essentially needs to be re-designed and re-coded from scratch for a new formal system, which can require an unreasonable amount of engineering."

It sounds like they've essentially tailored their approach for this one dataset, which limits the generality of their approach, a limitation that was not discussed in the paper.

There is also only one baseline considered, which renders their experimental findings rather weak. For these reasons, I think this work is not quite ready for publication at ICLR 2020, although future versions with stronger baselines and experiments could be quite impactful.